# The TIRS trial: Enrollment procedures and baseline characterization of a pediatric cohort to quantify the epidemiologic impact of targeted indoor residual spraying on *Aedes*-borne viruses in Merida, Mexico

James T. Earnest[1], Oscar D. Kirstein[2], Azael C. Mendoza[3], Gloria A. Barrera-Fuentes[3,4], Henry Puerta-Guardo[3,5], Manuel Parra-Cardeña[5], Kevin Yam-Trujillo[5], Matthew H. Collins[6], Norma Pavia-Ruz[4], Guadalupe Ayora-Talavera[5], Gabriela Gonzalez-Olvera[3], Anuar Medina-Barreiro[3], Wilberth Bibiano-Marin[3], Audrey Lenhart[7], M. Elizabeth Halloran[8,9], Ira Longini[10], Natalie Dean[11], Lance A. Waller[11], Amy M. Crisp[10], Fabian Correa-Morales[12], Jorge Palacio-Vargas[13], Pilar Granja-Perez[13], Salha Villanueva[13], Hugo Delfin-Gonzalez[3], Hector Gomez-Dantes[14], Pablo Manrique-Saide[3], Gonzalo M. Vazquez-Prokopec[1] *

1 Department of Environmental Sciences, Emory University, Atlanta, GA, United States of America, 2 Central Laboratory of Entomology and Parasitology, Ministry of Health, Jerusalem, Israel, 3 Unidad Colaborativa para Bioensayos Entomológicos, Campus de Ciencias Biológicas y Agropecuarias, Universidad Autónoma de Yucatán, Mérida, Yucatán, México, 4 Laboratorio de Hematología, Centro de Investigaciones Regionales Dr. Hideyo Noguchi, Universidad Autónoma de Yucatán, Mérida, Yucatán, México, 5 Laboratorio de Virología, Centro de Investigaciones Regionales Dr. Hideyo Noguchi, Universidad Autónoma de Yucatán, Mérida, Yucatán, México, 6 The Hope Clinic of the Emory Vaccine Center, Division of Infectious Diseases, Department of Medicine, Emory University School of Medicine, Decatur, GA, United States of America, 7 Entomology Branch, Division of Parasitic Diseases and Malaria, U.S. Centers for Disease Control and Prevention, Atlanta, GA, United States of America, 8 Vaccine and Infectious Disease Division, Fred Hutchinson Cancer Center, Seattle WA, United States of America, 9 Department of Biostatistics, University of Washington, Seattle WA, United States of America, 10 Department of Biostatistics, University of Florida, Gainesville, FL, United States of America, 11 Department of Biostatistics and Bioinformatics, Rollins School of Public Health, Emory University, Atlanta, GA, United States of America, 12 Centro Nacional de Programas Preventivos y Control de Enfermedades (CENAPRECE) Secretaría de Salud Mexico, Mexico City, Mexico, 13 Servicios de Salud de Yucatán, Merida, Yucatan, Mexico, 14 Health Systems Research Centre, National Institute of Public Health, Cuernavaca, México

☯ These authors contributed equally to this work.
* gmvazqu@emory.edu

## Abstract

*Aedes* mosquito-borne viruses (ABVs) place a substantial strain on public health resources in the Americas. Vector control of *Aedes* mosquitoes is an important public health strategy to decrease or prevent spread of ABVs. The ongoing Targeted Indoor Residual Spraying (TIRS) trial is an NIH-sponsored clinical trial to study the efficacy of a novel, proactive vector control technique to prevent dengue virus (DENV), Zika virus (ZIKV), and chikungunya virus (CHIKV) infections in the endemic city of Merida, Yucatan, Mexico. The primary outcome of the trial is laboratory-confirmed ABV infections in neighborhood clusters. Despite the difficulties caused by the COVID-19 pandemic, by early 2021 the TIRS trial completed enrollment of 4,792 children aged 2–15 years in 50 neighborhood clusters which were allocated to

**Data Availability Statement:** Deidentified individual participant data can be found at the Harvard Dataverse Network (https://doi.org/10.7910/DVN/GITKOQ).

**Funding:** This study is funded by the National Institutes of Health, National Institute of Allergy and Infectious Disease (U01AI148069; Vazquez-Prokopec, PI) and partially by grants from NIH/National Institute of General Medical Sciences (U54 GM111274; Halloran, PI), NIH/National Institute of Allergy and Infectious Disease (R37 AI0032042; Halloran, PI), and the Innovative Vector Control Consortium, IVCC (DFID:30041-105; Vazquez-Prokopec, PI). The funders have not had a role in the study's design and will not be involved in the collection, analysis, or interpretation of the data. MPD was supported by CONACyt. The findings and conclusions in this paper are those of the authors and do not necessarily represent the official position of the CDC.

**Competing interests:** The authors have declared that no competing interests exist.

control or intervention arms via a covariate-constrained randomization algorithm. Here, we describe the makeup and ABV seroprevalence of participants and mosquito population characteristics in both arms before TIRS administration. Baseline surveys showed similar distribution of age, sex, and socio-economic factors between the arms. Serum samples from 1,399 children were tested by commercially available ELISAs for presence of anti-ABV antibodies. We found that 45.1% of children were seropositive for one or more flaviviruses and 24.0% were seropositive for CHIKV. Of the flavivirus-positive participants, most were positive for ZIKV-neutralizing antibodies by focus reduction neutralization testing which indicated a higher proportion of participants with previous ZIKV than DENV infections within the cohort. Both study arms had statistically similar seroprevalence for all viruses tested, similar socio-demographic compositions, similar levels of *Ae. aegypti* infestation, and similar observed mosquito susceptibility to insecticides. These findings describe a population with a high rate of previous exposure to ZIKV and lower titers of neutralizing antibodies against DENV serotypes, suggesting susceptibility to future outbreaks of flaviviruses is possible, but proactive vector control may mitigate these risks.

## Introduction

*Aedes* mosquito-borne viruses (ABVs) such as dengue (DENV), Zika (ZIKV), and chikungunya (CHIKV) are among the leading cause of mosquito-borne illness in tropical urban environments. DENV is a flavivirus estimated to infect approximately 390 million people every year, causing an average of 2 million cases of severe illness leading to 21,000 deaths [1, 2]. DENV has become the most rapidly expanding vector-borne virus in the Americas, with Brazil, Colombia, and Mexico experiencing the most significant burden of disease in the region [3]. ZIKV, a flavivirus closely related to DENV, spread from Africa and Asia to the Americas in 2015, resulting in thousands of adverse fetal outcomes due to congenital infection. This epidemic was recognized in 2016 by the World Health Organization (WHO) as a Public Health Emergency of International Concern (PHEIC) [4]. The alphavirus CHIKV was first recognized in 1952–1953 in southeast Tanzania and northern Mozambique [5] and spread rapidly throughout Asia and the Americas, leading to explosive urban epidemics in the early 21st century causing significant morbidity [6–16]. While a vaccine against DENV exists, its usage is restricted to adolescents who have been previously exposed to DENV infection [17]. There are no approved treatments or vaccines for either ZIKV or CHIKV.

Control of *Aedes aegypti* is a vital strategy for containing ABV transmission in endemic regions [18]. Insecticides, typically ultra-low volume spraying alone or in combination with larval management and source reduction, remain the most commonly used tools to control ABVs [19]. However, recent studies have concluded that these control strategies have limited impact on ABV transmission [20] or lack sufficient information to conclude whether they effectively prevent transmission [21]. Due to the limitations of existing vector control tools, novel methods are being developed [18]. One of these techniques, targeted indoor residual spraying (TIRS), exploits the preference of *Ae. aegypti* to rest indoors and on surfaces below 1.5m and under or behind furniture to efficiently direct residual insecticide applications [22, 23]. TIRS is a cost-effective approach for insecticide delivery which is preventatively applied once a year before the ABV transmission season begins. TIRS is effective for up to 6 months and is applied to specific locations of mosquito landing in households [23]. Traditional

mosquito fogging procedures require frequent administration and are generally carried out reactively following reports of ABV transmission. Fogging is generally performed outdoors and in an indiscriminate manner while TIRS is applied indoors at the sites of mosquito resting within households. Because of this, TIRS is hypothesized to be more effective than outdoor spraying at controlling *Aedes* mosquitos and, thus, ABVs [24]. Evidence from entomological trials showed that TIRS significantly reduced *Ae. aegypti* density throughout the transmission season of Merida, Yucatan [23]. In Cairns, Australia, TIRS (combined with contact tracing) led to an 89% reduction in DENV cases during an outbreak in 2009 [25]. While both application modes of TIRS are compatible (preventive application versus outbreak response), preventive application has gained more interest due to its ability to impact mosquitoes over an entire transmission season, which mathematical models suggest will lead to the biggest impact in DENV transmission [24, 26, 27].

Current WHO guidelines for vector control strategies require evidence from field trials with epidemiological endpoints to inform policy recommendations [28]. In response to this need, a cluster randomized controlled trial (CRCT) quantifying the efficacy of TIRS applications in preventing ABV infections began in 2020 in Merida [22, 29]. The TIRS trial includes several innovations in the implementation of ABV CRCTs. First, we restricted the trial region to previously identified ABV-transmission hotspots [30] which reduced the number of clusters required [22]. Second, we developed an expanded covariate-constrained randomization algorithm to subsample 50 census tracts from within these hotspots, ensuring balance across several cluster-level covariates and maximizing the distance between clusters to limit shared boundaries. Third, to minimize contamination from adjacent neighborhood blocks due to mosquito movement, we used a"fried egg" design in which the intervention is applied within clusters measuring 5x5 city blocks but the primary endpoint is measured in the "yolks", i.e., the central 3x3 [22, 31]. Finally, we capture human mobility data in the analysis to minimize contamination due to people spending time outside of treated houses [32]. This report conveys our analysis of baseline demographic, socioeconomic, and seroprevalence data, blinded to arm allocation, for the TIRS trial running from 2020–2024.

## Materials and methods

### Statement of ethics

The trial protocol was approved by Emory University (IRB00108666) and the Autonomous University of Yucatan (CEI-05-2020) and received endorsement by the Ministry of Health (MOH) of the State of Yucatan and the Federal Ministry of Health of Mexico (CENAPRECE). Written consent/assent was obtained from the parents of all adolescent participants (aged 2–15 years at time of enrollment) in their native Spanish language. The trial was registered in ClinicalTrials.gov (identifier NCT04343521; registered on April 13, 2020) and complied with the WHO International Clinical Trials Registry Platform requirements (ICTRP) (see S1 Checklist). An independent external monitor reviews consents, adverse effects, and other trial performance metrics yearly.

### Study site

Merida, the capital of the Mexican State of Yucatan, has a population of 892,000 [33]. The climate is tropical, with a mean annual temperature of 25.9˚C and seasonal precipitation from July-December. ABVs in Merida are transmitted in a highly seasonal pattern coinciding with the rainy season [30, 34]. The ABV transmission season (occurring from July to November) and incidence rate in Merida have been previously characterized [34–36] and are consistent with high levels of transmission among the population (e.g., in 2015, the incidence of

symptomatic ABVs was 8.6 cases per 1,000 person-years for CHIKV, 3.5 cases per 1,000 person-years for DENV, and 2.3 per 1,000 person-years for ZIKV [34–36]). *Aedes aegypti* mosquitoes infest houses in high numbers [37] and are highly resistant to pyrethroid insecticides [38, 39]. Emory University and the Collaborative Unit for Entomological Bioassays (UCBE) from the Autonomous University of Yucatan are the implementation partners for the TIRS trial responsible for performing entomological field work within the city [22]. UCBE is a reference laboratory for the State and Federal Ministries of Health and a WHO Good Laboratory Practice (GLP) site for evaluating insecticide products for vector control [40]. The Autonomous University of Yucatan and Yucatan State Laboratory for Public Health (SSY) were implementation partners for the serological analyses of samples taken at baseline.

## Study design

The TIRS trial is an ongoing two-arm, parallel, unblinded, cluster randomized controlled trial conducted in 50 clusters of 5x5 city blocks each (1:1 allocation) within Merida [22]. For the control arm, routine vector control actions are carried out in response to reports of symptomatic ABV cases. For the intervention arm, routine vector control is carried out in addition to TIRS. The primary endpoint of the trial is to evaluate the effectiveness of TIRS by quantifying the incidence of ABV-confirmed clinical cases within the cohorts. A secondary epidemiological endpoint is to measure seroconversion rates against one or more ABVs observed from annual blood draws from the cohort participants (for example, a participant with a negative result at baseline becomes positive in the years after baseline analysis). Entomological secondary endpoints include quantifying the impact of TIRS on indoor *Ae. aegypti* density and on ABV infection in female *Ae. aegypti* [22]. The study was designed to have 80% power in detecting a minimum of 92 events, which will lead to quantify an effect size of 70% for the TIRS intervention [22]. More information about the design of the TIRS trial [22] and the covariate-constrained randomization procedure [29] are described in previous publications.

## Participant enrollment and impact of COVID-19

The TIRS trial began during the first wave of SARS-CoV-2 transmission in Merida on October 1st, 2020 [41]. Consequently, strategies for training of the TIRS staff, household visits, participant engagement, enrollment, and specimen collections were modified to account for existing social distancing norms, field worker protection, and participant safety. One of the first modifications was the remote training of the TIRS trial staff, which included virtual sessions explaining the trial procedures. There were small (10–15) group meetings to discuss strategies and provide PPE for field use (acrylic face shield, N95 face masks, gloves, and a mat with disinfectant to be used before entering every household when the meetings occurred indoors). Trial enrollment concluded on January 24th, 2022.

Enrollment activities were modified to include the implementation of remote (non-participatory) activities to inform participants within a cluster that they were part of the trial, such as: a) a field truck with project logos on the door broadcasting an audio message from the street; b) printed materials left at every house with information about the trial and links to the project website (https://beta.clinicaltrials.gov/study/NCT04343521); and c) house-to-house visits by project staff. Thirty enrollment personnel were deployed into teams consisting of one nurse and one anthropologist/social scientist. These teams traveled to the field to enroll children and houses in 50 neighborhood clusters. This allowed for more direct interaction with study participants in hopes of encouraging higher rates of acceptance and engagement.

After informing participants of the study and giving them enough time to consider participating in the trial, epidemiological field teams contacted every household with children aged

2–15 years and scheduled a time to perform informed consent/assent. Consent/assent was mandatory for participating in the study, and parents were given the choice to conduct it either indoors or outside the house. The report of any family member having COVID-19 symptoms prompted our team to reschedule enrollment after two weeks of anybody in the house showing symptoms. While the initial plan for enrollment (before COVID-19) was to enroll all individuals during October-December of 2020 and conduct baseline serology during the inter-epidemic period from January-March of 2021 [22], delays in enrollment and the need to minimize contact with participants prompted the team to conduct both enrollment and baseline blood specimen collections at the same time (after consent/assent, participants provided a blood specimen for baseline ABV characterization). At this point, additional sources of information about the house, children's mobility behavior, and the family's socio-demographics were collected. Therefore, enrollment occurred during a period in which Merida was subject to closing of high-density social locations (bars, restaurants, churches), mobility restrictions by the general public, and the 'safe distance' program involving wearing face masks and limiting distance between people [41].

## Baseline sources of information

**Socioeconomic and demographic data collection.** Household socio-economic and demographic characteristics were registered for all houses with enrolled children. Briefly, questions about the number of residents (and their ages), length of time the family has occupied the house, the size and building materials of the house, sanitation and technology used in the house, and knowledge and attitudes about ABVs were asked at enrollment.

**Routine mobility patterns.** Given the role of human mobility in shaping DENV transmission [42], and that the trial will include mobility information in a secondary analysis [22], we characterized routine mobility patterns of enrolled children at baseline. Briefly, we performed a semi-structured interview that aimed to quantify the time participants spend at home each day of the week. In a second section, the mobility survey asked participants about the type and number of locations they visit when they leave their house. This mobility survey has been extensively used and validated with GPS data In Iquitos, Peru and further information has been published previously [42, 43].

**Specimen collection and IgG seroprevalence of cohort participants.** Baseline blood samples were collected from all participants living in the central "yolk" of the clusters and an equal number of participants residing in the periphery via standard phlebotomy according to the TIRS protocol [22]. Vacutainer tubes (BD cat# 368159) with blood samples were kept in coolers with cold blocks in the field for no more than 2 hours. Samples were transported to the project's lab for centrifugation, and serum was aliquoted and stored at -80˚C.

Immunoglobulin G (IgG) ELISA tests were run in the SSY which has been certified by the Mexican MOH to conduct laboratory diagnostics of ABV cases. For DENV, a PanBio Dengue IgG Indirect ELISA (Cat# 01PE30, 96 Well/Kit, Abbott, Chicago, IL, USA) was used following the manufacturer's instructions with standard cutoff points for defining positive ($\geq$ 12 Panbio units) and negative samples ($<$ 9 Panbio units). Samples reading between these cutoff thresholds were classified as "indetermined". For ZIKV and CHIKV, we tested samples by commercial anti-ZIKV (Anti-ZIKV Virus ELISA IgG, Cat# EI 2668-9601G, Euroimmun, Germany) and anti-CHIKV (Anti-Chikungunya Virus IgG ELISA, Cat# EI2668-9601G, Euroimmun, Germany) IgG ELISAs according to the manufacturer's protocol. Serostatus in both assays was defined by a ratio of sample OD to the OD of a cutoff control as negative (Ratio $<$0.8), indetermined (Ratio 0.8–1.1), or positive (Ratio $>$1.1). The performance of all three commercial ELISAs used to detect the presence of IgG against DENV, ZIKV, and CHIKV has been evaluated

in previous studies [30, 36, 44, 45]. All samples were run using a fully automated spectrophotometer microplate reader (Thunderbolt Analyzer, Gold Standard Diagnostics).

*Identification of prior flavivirus infection by focus reduction neutralization tests (FRNT).*
Infection by one orthoflavivirus commonly induces antibodies that cross-react with one or more heterologous viruses in the genus, which causes ELISA assays to have low specificity. This is particularly relevant in places like Merida, where multiple flaviviruses are endemic or have previously circulated [30]. Thus, we considered a positive result on DENV or ZIKV IgG ELISA to indicate prior flavivirus infection and analyzed relative titers of neutralizing antibodies determined by FRNT to distinguish between the two viruses [46]. To determine neutralizing antibody titers ($FRNT_{50}$, the inverse of the serum dilution that exhibits 50% of maximum neutralization) [47] against DENV1-4 and Zika, 100 FFU of ZIKA (strain: MR766), DENV-1 (strain: West Pac 74), DENV-2 (strain: S-16803), DENV-3 (strain: ARB-1669/2019), or DENV-4 (strain: TVP-360) was incubated with 3-fold dilutions (dilution factors: 1:20–1:14,580) of participant serum for one hour at 37°C. The virus-serum mixtures were inoculated onto Vero cells and infection was allowed to proceed for 1h at 37°C. Cells were overlaid with 2.5% carboxymethylcellulose (Sigma cat# C5013) in DMEM and incubated at 37°C for 72h. Plates were fixed with 2% paraformaldehyde (Sigma cat# P6148) and cells were permeabilized with 0.1% saponin (Sigma cat# 47036) in phosphate buffered saline with 0.1% bovine serum albumin (BSA; Sigma cat# A9418) and 1% fetal bovine serum (Corning cat# 45001–106) for blocking. Intracellular virus protein was detected with a mouse monoclonal antibody that is cross-reactive to multiple orthoflaviviruses (4G2 diluted at 1μg/ml in permeabilization buffer), which was detected with anti-mouse HRP (Jackson Laboratories cat# AB_10015289) diluted 1:2000 in permeabilization buffer and KBL TrueBlue substrate (SeraCare cat# 5510–0030). Foci were counted on a BioReader 7000E spot counting machine (BIOSYS). $FRNT_{50}$ was interpreted as below our assay's detection limit if <50% neutralizing activity was calculated as ≤30 and titers with 50% neutralizing activity at the final dilution (14,580) were considered at the assay's maximum and reported as $FRNT_{50} = 14,580$. The identity of flavivirus serotypes that have previously infected each participant were inferred by comparing relative $FRNT_{50}$ values for each virus, with a ≥ 4-fold difference in the $FRNT_{50}$ considered a significant difference [47]. A participant with ≥4-fold higher $FRNT_{50}$ against a single virus was considered previously exposed to that virus. Participants with $FRNT_{50}$ values within a 4-fold difference were considered "multitypic".

*Aedes aegypti collections.* We conducted a baseline entomological quantification of *Ae. aegypti* density to check whether the randomization procedure led to a balanced distribution of mosquitoes between arms. Two entomological indices were calculated: a) indoor *Ae. aegypti* adult density, based on a random sample of 30 houses per cluster located in the "yolk" of each cluster (totaling 1,500 houses, split 1:1 between arms) using Prokopack aspirator collections performed for 10 min per house [39, 48]; b) Outdoor number of *Ae. aegypti* eggs (estimated from a network of >5,000 ovitraps deployed throughout the city of Merida by the Yucatan MOH [48]). Ovitraps were regularly monitored by the Yucatan MOH within each study cluster and were selected to query the national database for information on the weekly number of eggs from 2015 to 2019 (the last five years before the trial) in order to statistically compare eggs counts between study arms. Open-source QGIS software [49] was used to monitor the spatial distribution of ovitraps.

*Aedes aegypti susceptibility to insecticides.* The Centers for Disease Control and Prevention (CDC) bottle bioassay [50] was used to test for susceptibility to the active ingredient used in the TIRS trial (pirimiphos-methyl) as well as other commonly used insecticides (malathion, deltamethrin, and permethrin). The diagnostic doses and times were used as recommended by the CDC guidelines, including a dose of 25 μg/bottle for pirimiphos-methyl. The baseline

susceptibility screening included *Ae. aegypti* mosquitoes collected from 16 randomly selected clusters, ten from the arm receiving the TIRS treatment and six from the control arm.

## Statistical analyses

All data was stored on secure data servers and participant identifiers were blinded using non-identifiable codes based on their households. Households were assigned codes unique to the project database and the identity of these households as "control" or "intervention" was withheld from those doing data analysis. Here, we will report trial arms as "A" and "B" to designate their treatment allocation. Descriptive tables were generated for each study arm and analyzed using chi$^2$ tests of independence. Seroprevalence rates were compared between arms using Generalized Linear Mixed models (GLMM), considering arm as a fixed effect and cluster ID as a random intercept. Entomological measures (ovitrap and indoor adult collections) were analyzed using negative-binomial GLMMs that included arm as a fixed effect and cluster ID as a random intercept. All analyses were performed within the R programming environment (https://www.r-project.org/; version 4.3.1), and GLMMs were run using the *lme4* package [51]. Deidentified individual participant data can be found at the Harvard Dataverse Network (https://doi.org/10.7910/DVN/GITKOQ).

## Results and discussion

### Participant enrollment

Following all the procedures to minimize COVID-19 exposure, the trial successfully enrolled 4,792 children. **Fig 1** shows the distribution of clusters in which children were enrolled. More children were enrolled than initially proposed due to 230 children who moved from the study area soon after the enrollment process (**S1 Fig**). This loss of participants was much higher than in any of our prior studies [34–36] and is potentially related to the effects of COVID-19. Following additional loss of participants, 4,461 children from the 50 randomized neighborhood clusters remained in the study, with 3,542 (79.4%) providing a serum sample for serological testing. From this cohort, 1,399 participant serum samples from children living in their cluster's "yolk" blocks were randomly selected and tested for IgG seroreactivity to ABVs.

### Demographic and socio-economic characteristics

To ensure that both arms consisted of a population with a similar demographic and socio-economic profile, we surveyed participants at enrollment for these characteristics. Both arms had a similar distribution of male and female participants with overall identity of 51.6% and 48.4%, respectively and similar distributions of ages among participants (**Table 1**). We found that both arms of the TIRS study had similar responses to questionnaires concerning socioeconomic and demographic characteristics (**Table 2**). The majority of participants lived in households with 4–6 total members (63.3%) consisting of 1 or 2 children in the trial's enrollment age range (2–16 years old) (42.0% and 36.2%, respectively). The participants' houses typically contained 3–4 or 5–6 rooms (both groups with 39.2%) and 92.1% of houses were a single story. More than 90% of the houses had a backyard, and 85.7% had a front yard. Similarly, house construction materials were uniform; 98.0% of the participants lived in houses with concrete walls, 86.3% lived in houses with concrete roofs and 10.8% lived in a house with a mixed concrete/metal roof. The floors were the only construction material that varied among participating households. The most common floor type was concrete, with 35.4% of participants having concrete floors. Brick and ceramic floors were the next most common, with 23.3% and 23.8% of participants having these, respectively. Most houses had 3–5 (42.2%) or 6–8 (34.7%)

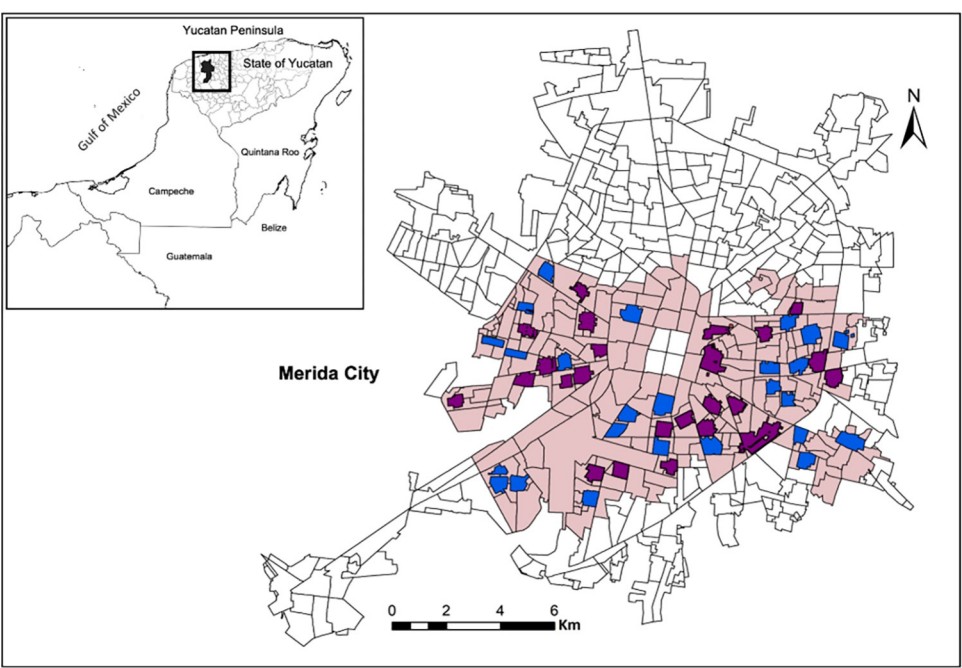

**Fig 1. Distribution of blinded control and intervention clusters.** A map of Merida, Yucatan, Mexico with blinded control and intervention clusters (Arm A clusters in purple and Arm B clusters in blue) The pink coloring shows the ABV hotspot area identified for Merida [52]. The source of the census tract boundaries was the Instituto de Estadistica y Geografia (INEGI), 2010.

windows. A similar percentage of houses had 1–2 (48.7%) doors or 3–5 (40.1%) doors, with only 11.1% having more than 5. An approximately equal percentage of households had screened windows and doors (35.1%), unscreened windows and doors (34.9%), or windows and doors with broken screens (27.8%).

We measured several socioeconomic factors that might contribute to disease exposure due to resource or information availability (**Table 2**). Participating households overwhelmingly had access to potable water (98.7%), with 94.4% coming from municipal sources. Similarly, 97.9% of participants had access to electricity. Access to computers was less common as we observed only 51.7% of households owned a computer. However, due to access to smartphones and other means, 96.4% of households reported having access to the internet. We measured

**Table 1. Demographics of study participants enrolled in the control or intervention arms of the TIRS trial.** Control and intervention arms are blinded as Arm A or Arm B.

| | Arm A | | Arm B | | Total | |
|---|---|---|---|---|---|---|
| **Sex** | **N** | **%** | **N** | **%** | **N** | **%** |
| Female | 1053 | 47.4 | 1108 | 49.5 | 2161 | 48.4 |
| Male | 1169 | 52.6 | 1131 | 50.5 | 2300 | 51.6 |
| Total | 2222 | 100.0 | 2239 | 100.0 | 4461 | 100.0 |
| **Age at Enrollment** | | | | | | |
| 2–5 | 621 | 27.9 | 599 | 26.8 | 1220 | 27.3 |
| 6–8 | 512 | 23.0 | 539 | 24.1 | 1051 | 23.6 |
| 9–12 | 662 | 29.8 | 684 | 30.5 | 1346 | 30.2 |
| 13–15 | 427 | 19.2 | 417 | 18.6 | 844 | 18.9 |
| Total | 2222 | 100.0 | 2239 | 100.0 | 4461 | 100.0 |

**Table 2. Common socio-economic factors describing the households enrolled in the TIRS trial.** The control and intervention arms were blinded as either Arm A or Arm B.

| | Arm A | | Arm B | | Total | |
|---|---|---|---|---|---|---|
| | N | % | N | % | N | % |
| **Total Number of People in Household** | | | | | | |
| 1–3 | 68 | 14.4 | 83 | 16.8 | 151 | 15.6 |
| 4–6 | 301 | 63.9 | 310 | 62.6 | 611 | 63.3 |
| 7–9 | 83 | 17.6 | 83 | 16.8 | 166 | 17.2 |
| >9 | 18 | 3.8 | 19 | 3.8 | 37 | 3.8 |
| **Total Number of Children in Household** | | | | | | |
| 1 | 187 | 39.7 | 219 | 44.2 | 406 | 42.0 |
| 2 | 178 | 37.8 | 171 | 34.5 | 349 | 36.1 |
| 3 | 66 | 14.0 | 71 | 14.3 | 137 | 14.2 |
| 4 | 30 | 6.4 | 18 | 3.6 | 48 | 5.0 |
| >4 | 9 | 1.9 | 16 | 3.2 | 25 | 2.6 |
| **Number of Rooms in House** | | | | | | |
| 1–2 | 42 | 8.9 | 41 | 8.3 | 83 | 8.6 |
| 3–4 | 194 | 41.2 | 185 | 37.4 | 379 | 39.2 |
| 5–6 | 172 | 36.5 | 207 | 41.8 | 379 | 39.2 |
| >6 | 62 | 13.2 | 62 | 12.5 | 124 | 12.8 |
| **Number of floors/levels** | | | | | | |
| 1 | 427 | 90.7 | 463 | 93.5 | 890 | 92.1 |
| >1 | 43 | 9.1 | 31 | 6.3 | 74 | 7.7 |
| **Backyard** | | | | | | |
| Yes | 428 | 90.9 | 470 | 94.9 | 898 | 93.0 |
| No | 42 | 8.9 | 25 | 5.1 | 67 | 6.9 |
| **Roof** | | | | | | |
| Concrete | 406 | 86.2 | 428 | 86.5 | 834 | 86.3 |
| Cardboard | 1 | 0.2 | 3 | 0.6 | 4 | 0.4 |
| Metal | 12 | 2.5 | 9 | 1.8 | 21 | 2.2 |
| Thatch | 0 | 0.0 | 0 | 0.0 | 0 | 0.0 |
| Mixed | 50 | 10.6 | 54 | 10.9 | 104 | 10.8 |
| Other | 1 | 0.2 | 0 | 0.0 | 1 | 0.1 |
| **Floor** | | | | | | |
| Concrete | 178 | 37.8 | 164 | 33.1 | 342 | 35.4 |
| Brick | 89 | 18.9 | 136 | 27.5 | 225 | 23.3 |
| Ceramic | 128 | 27.2 | 102 | 20.6 | 230 | 23.8 |
| Soil | 1 | 0.2 | 0 | 0.0 | 1 | 0.1 |
| Mixed | 74 | 15.7 | 87 | 17.6 | 161 | 16.7 |
| Other | 0 | 0.0 | 5 | 1.0 | 5 | 0.5 |
| **Walls** | | | | | | |
| Concrete | 460 | 97.7 | 487 | 98.4 | 947 | 98.0 |
| Cardboard | 1 | 0.2 | 2 | 0.4 | 3 | 0.3 |
| Metal | 2 | 0.4 | 1 | 0.2 | 3 | 0.3 |
| Thatch | 0 | 0.0 | 0 | 0.0 | 0 | 0.0 |
| Mixed | 6 | 1.3 | 2 | 0.4 | 8 | 0.8 |
| Brick | 0 | 0.0 | 0 | 0.0 | 0 | 0.0 |
| Other | 1 | 0.2 | 1 | 0.2 | 2 | 0.2 |
| **Screened Doors/Windows** | | | | | | |

*(Continued)*

**Table 2.** (Continued)

|  | Arm A | | Arm B | | Total | |
|---|---|---|---|---|---|---|
|  | N | % | N | % | N | % |
| Yes | 162 | 34.4 | 177 | 35.8 | 339 | 35.1 |
| Broken | 134 | 28.5 | 135 | 27.3 | 269 | 27.8 |
| No | 174 | 36.9 | 163 | 32.9 | 337 | 34.9 |
| **Water Supply** |  |  |  |  |  |  |
| Running water | 442 | 93.8 | 470 | 94.9 | 912 | 94.4 |
| Water Tank | 7 | 1.5 | 4 | 0.8 | 11 | 1.1 |
| Well | 11 | 2.3 | 9 | 1.8 | 20 | 2.1 |
| Manual Pump | 0 | 0.0 | 0 | 0.0 | 0 | 0.0 |
| Other | 1 | 0.2 | 5 | 1.0 | 6 | 0.6 |
| **Electricity** |  |  |  |  |  |  |
| Yes | 463 | 98.3 | 483 | 97.6 | 946 | 97.9 |
| No | 7 | 1.5 | 11 | 2.2 | 18 | 1.9 |
| **Owns a Computer** |  |  |  |  |  |  |
| Yes | 251 | 53.3 | 248 | 50.1 | 499 | 51.7 |
| No | 218 | 46.3 | 247 | 49.9 | 465 | 48.1 |
| **Has Internet Access** |  |  |  |  |  |  |
| Yes | 454 | 96.4 | 477 | 96.4 | 931 | 96.4 |
| No | 10 | 2.1 | 12 | 2.4 | 22 | 2.3 |
| **Use Insect Repellents** |  |  |  |  |  |  |
| All Day | 32 | 6.8 | 44 | 8.9 | 76 | 7.9 |
| Morning | 22 | 4.7 | 22 | 4.4 | 44 | 4.6 |
| Evening | 71 | 15.1 | 68 | 13.7 | 139 | 14.4 |
| Night | 171 | 36.3 | 150 | 30.3 | 321 | 33.2 |
| Evening and Night | 27 | 5.7 | 30 | 6.1 | 57 | 5.9 |
| No | 148 | 31.4 | 181 | 36.6 | 329 | 34.1 |
| **Use Insecticides** |  |  |  |  |  |  |
| Yes | 353 | 74.9 | 383 | 77.4 | 736 | 76.2 |
| No | 117 | 24.8 | 111 | 22.4 | 228 | 23.6 |

the use and implementation of insect avoidance techniques within the cohort and found 34.1% of participating families reported no use of insect repellent. Among those that did, the plurality (33.2%) used repellents only at night. Similarly, 23.7% of participants did not use any insecticides in their households.

## Mobility surveys

A total of 3,881 trial participants provided routine mobility surveys. The baseline survey took place during stay-at-home and virtual learning mandates imposed by Mexico's MOH in response to COVID-19 [41], which explains why over 75% of participants reported staying at home 90% or more of the time (**Fig 2A**). This behavior was similar across both study arms (**Fig 2A**) and throughout the week, with weekends evidencing a slightly lower presence at home than weekdays (**Fig 2B**). Less than half of the participants (1,682/4,015, 41.9%) reported visiting at least one out-of-home location, which involved primarily relatives (84%) but also friends (4.5%), their parent's workplace (2.9%), businesses (4.6%), parks (1.2%), and daycare or private schools (1.0%). The low frequency of school visits in our pediatric population was because Merida imposed virtual learning sessions throughout the first wave of the pandemic

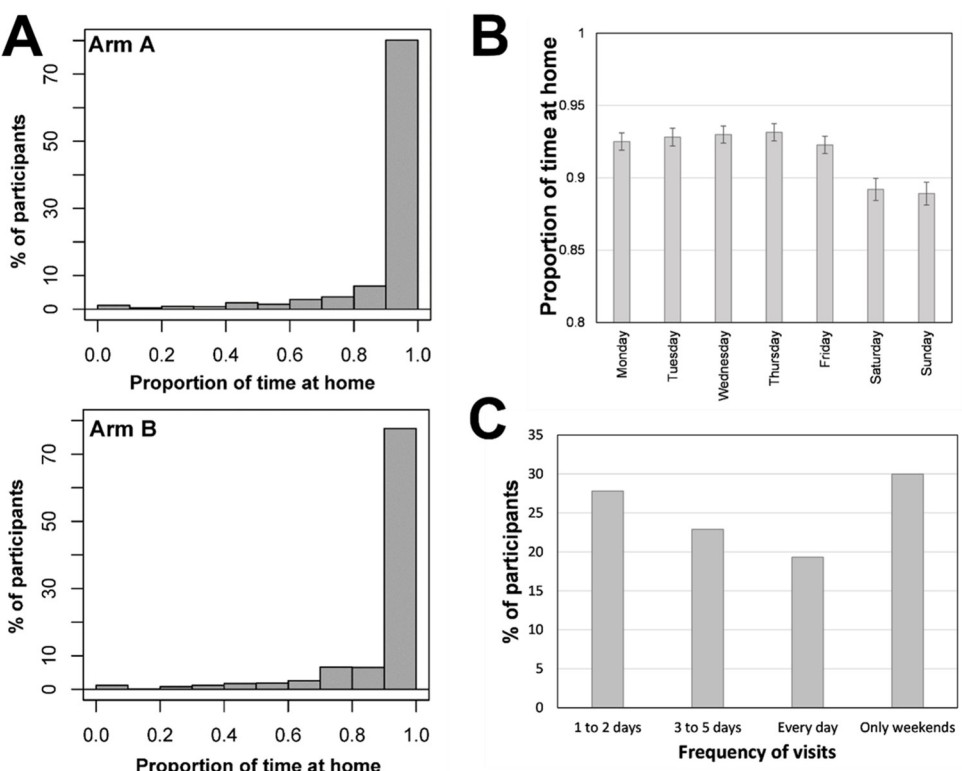

**Fig 2. Summary of prospective mobility patterns of study participants in the TIRS trial.** (A) The proportion of time participants enrolled in each study arm spent at their house (from a 24h day). (B) The time all participants spent at home, separated by day of the week. (C) The frequency of visits (in times per week) to places other than their home.

[41]. **Fig 2C** shows that the visitation frequency of those out-of-home locations was equally distributed across the specified times of the week.

## Baseline seroprevalence of IgG against ABVs

Flavivirus seroprevalence was defined by reactivity to DENV and/or ZIKV in IgG ELISA tests. We observed that 45.1% (N = 631) of participants were flavivirus seropositive (**Table 3**). Seroprevalence was similar between arms and between female and male participants (**Table 3**).

**Table 3. Seroprevalence of IgG against flavivirus (DENV and/or ZIKV) and CHIKV, stratified by sex and age at TIRS baseline sampling.**

| | Arm A | | | | Arm B | | | | Total | | | |
|---|---|---|---|---|---|---|---|---|---|---|---|---|
| | Flavivirus | | CHIKV | | Flavivirus | | CHIKV | | Flavivirus | | CHIKV | |
| Sex | N | Seroprevalence (%) | N | Seroprevalence (%) | N | Seroprevalence (%) | N | Seroprevalence (%) | N | Seroprevalence (%) | N | Seroprevalence (%) |
| Female | 148 | 45.0 | 89 | 27.1 | 177 | 47.2 | 85 | 22.7 | 325 | 46.1 | 174 | 24.7 |
| Male | 163 | 44.1 | 77 | 21.0 | 143 | 44.0 | 85 | 26.1 | 306 | 44.1 | 162 | 23.3 |
| Total | 311 | 44.5 | 166 | 23.7 | 320 | 45.7 | 170 | 24.3 | 631 | 45.1 | 336 | 24.0 |
| **Age** | | | | | | | | | | | | |
| 2–5 | 21 | 12.8 | 9 | 5.5 | 21 | 15.0 | 7 | 5.0 | 42 | 13.9 | 16 | 5.3 |
| 6–8 | 46 | 29.1 | 32 | 20.2 | 66 | 43.7 | 28 | 18.5 | 112 | 36.2 | 60 | 19.4 |
| 9–12 | 113 | 55.4 | 61 | 29.9 | 116 | 48.5 | 69 | 28.9 | 229 | 51.7 | 130 | 29.3 |
| 13–16 | 131 | 75.3 | 64 | 36.8 | 117 | 68.9 | 66 | 38.8 | 248 | 72.1 | 130 | 37.8 |

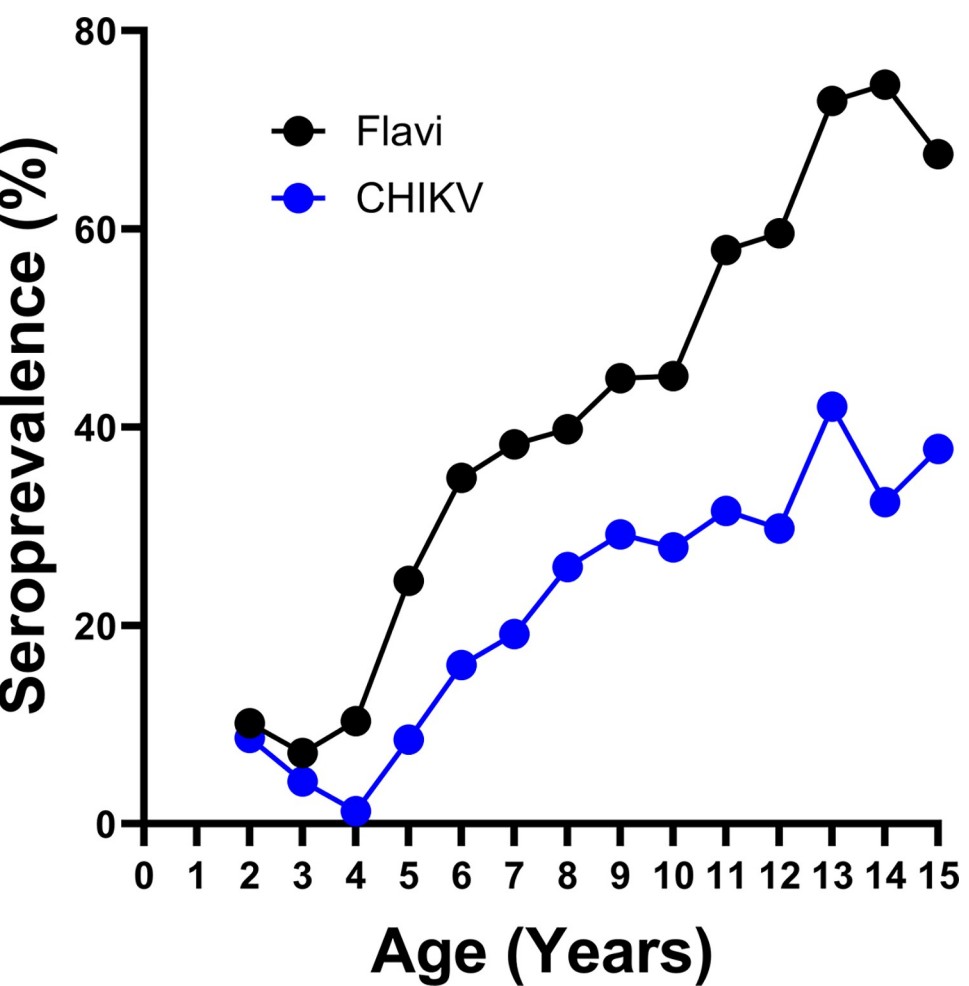

**Fig 3. Overall seroprevalence of participants by age.** Seroprevalence (%) or participants with antibodies against flavivirus (black) and CHIKV (blue) at the indicated age at time of sampling.

Seroprevalence to ABVs in our cohort increased with age at time of baseline sampling, as is commonly observed due to cumulative infections over time in endemic areas (**Table 3** and **Fig 3**). For flavivirus antibodies, the increase ranged from 7.1% in the 3—year-old participants to 74.6% in 14-year-old participants (**Fig 3**). We observed a similar trend for CHIKV, with overall seroprevalence of 24.0% (**Table 3**), increasing from 1.3% in the 4-year-old participants to 42.1% in the 13-year-old participants (**Fig 3**), suggesting endemic transmission of CHIKV among the population of Merida. Of the 336 CHIKV seropositive participants, we observed that 221 (65.7%) were also seropositive for flavivirus antibodies.

While we observed 631 flavivirus reactive participants by IgG ELISA, we were unable to attribute the result to a specific flavivirus due to the inherent cross-reactive nature of anti-ZIKV and anti-DENV polyclonal antibodies [45]. To classify previous exposure more precisely, we performed FRNT assays with all flavivirus-positive sera against ZIKV and DENV serotypes 1–4. The results indicated a wide range of neutralizing antibody titers from the lower ($FRNT_{50}$ = 30) and upper ($FRNT_{50}$ = 14,580) limits of detection for the assay (**Fig 4A**). We saw a trend of higher $FRNT_{50}$ titers among positive samples against ZIKV than DENV-1, DENV-2, DENV-3, and DENV-4 (median $FRNT_{50}$ = 1466, 739, 548, 61, and 114, respectively)

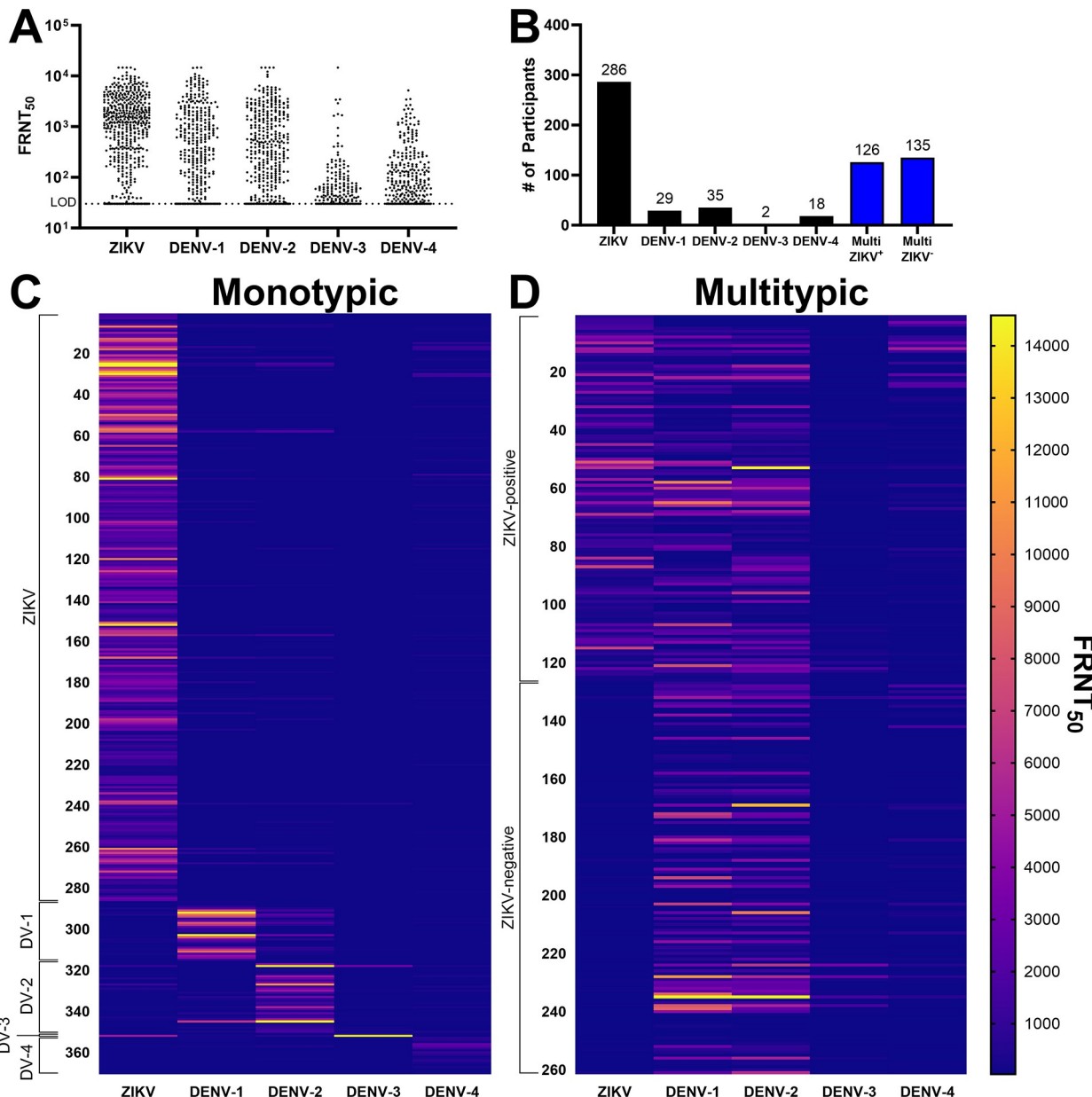

**Fig 4. Flavivirus exposure history of trial participants.** (**A**) Calculated $FRNT_{50}$ titers of 631 flavivirus-reactive participants were determined for ZIKV and DENV serotypes 1–4. (**B**) Participants with $FRNT_{50}$ titers ≥4-fold higher against any one serotype were considered to be exposed to a monotypic infection while those with titers to two or more serotypes with <4-fold difference were considered multitypic. Multitypic participants were split into ZIKV-exposed (Multi ZIKV+) or unexposed (Multi ZIKV-) based on whether ZIKV titers were within 4-fold of any DENV serotype titer. (**C-D**) Heatmaps of $FRNT_{50}$ titers against ZIKV and DENV 1–4 for all participants.

suggesting the antibody response to ZIKV may be more neutralizing than those against DENV within this cohort due to recency of infection or other unknown factors.

The identity of previous ZIKV and/or DENV infection was determined by comparing $FRNT_{50}$ titers of individual participants between ZIKV and the four DENV serotypes. We observed indications of previous monotypic infection with ZIKV or DENV in 370 participants (**Fig 4B** and **S2A Fig**), with the majority of these indicating prior ZIKV infection alone (N = 286). Participants with titers against more than one flavivirus or more than one DENV

**Table 4. Results from negative-binomial GLMMs analyzing the difference between the blinded control and intervention arms (either Arm A or Arm B) for, a) the total number of *Ae. aegypti* per house (males and females), and b) the number of female *Ae. aegypti* per house.**

| Entomological Index | Fixed effect | Estimate | Std. Error | Z-value | P-value |
|---|---|---|---|---|---|
| a) Total *Ae. aegypti* | Intercept | -1.179 | 0.095 | -12.390 | <0.001 |
| | Arm | -0.097 | 0.109 | -0.887 | 0.375 |
| b) *Ae. aegypti* females | Intercept | -1.936 | 0.126 | -15.382 | <0.001 |
| | Arm | 0.026 | 0.129 | 0.200 | 0.841 |

serotype that were within 4-fold of each other were considered multitypic (**Fig 4B**, blue), indicating either previous infections by two or more serotypes or high levels of cross-reactive antibodies. Of the multitypic flavivirus-immune participants, 48% had cross-reactive antibodies to ZIKV and DENV, while 52% cross-reacted within DENV serotypes, but were negative for ZIKV (**Fig 4C and 4D**). 98.6% of the ZIKV-reactive participants were over the age of 5 at the time of sampling in 2021 (**S2B Fig**) meaning they were alive at the beginning the 2016 ZIKV pandemic. However, we did observe 4 ZIKV-positive, DENV-negative participants at 4 years of age or younger, suggesting potential exposure to ZIKV after 2016.

## Baseline entomology

The indoor relative adult *Ae. aegypti* density (±SD) at baseline averaged 0.78 (±1.7) per house in Arm A and 0.82 (±1.66) per house in Arm B, whereas relative female *Ae. aegypti* density averaged 0.45 (±1.1) and 0.41 (±0.94), respectively. For both indices, a negative binomial GLMM found no statistical difference between arms (**Table 4**). To further explore whether the study arms were balanced with regards to *Ae. aegypti* density, we calculated the mean number of eggs per ovitrap. We compared this between arms for the five years before the beginning of the trial (2015–2019), finding that ovitrap data never reflected an imbalance in the mean number of eggs between arms (**Fig 5**).

## Susceptibility to insecticides

CDC bottle bioassays showed no evidence of resistance to pirimiphos-methyl at baseline in the 10 clusters assigned to receive TIRS (mean mortality, 99.5%, range 98–100%) (**Fig 6**). For the six control clusters, mean mortality was 97.8% (range 94–100%). Furthermore, *Ae. aegypti* from both treatment arms similarly showed full susceptibility to malathion (mean mortality, 100%), the emergence of resistance for deltamethrin (93.4% mortality in treatment clusters and 98% mortality in control clusters), and full resistance to permethrin (mean mortality of 58.1% and 59.4% in treatment and control clusters, respectively).

## Conclusions

The TIRS trial aims to quantify the epidemiological impact of a novel urban mosquito control method in preventing ABV-mediated diseases. Our baseline assessment suggests that, prior to the beginning of the trial, both study arms were balanced concerning the demographic characteristics of enrolled children, ABV seroprevalence, *Ae. aegypti* density, and susceptibility of *Ae. aegypti* to the insecticide active ingredient to be employed during the trial. Our serological results indicated that 45.1% of the cohort had flavivirus reactive antibodies in their serum at baseline, and 24% had CHIKV-reactive antibodies. A further analysis of the identity of flavivirus infections provides a thorough clinical description of a pediatric population living in an ABV-endemic area. Our findings indicated a large proportion of the cohort was likely

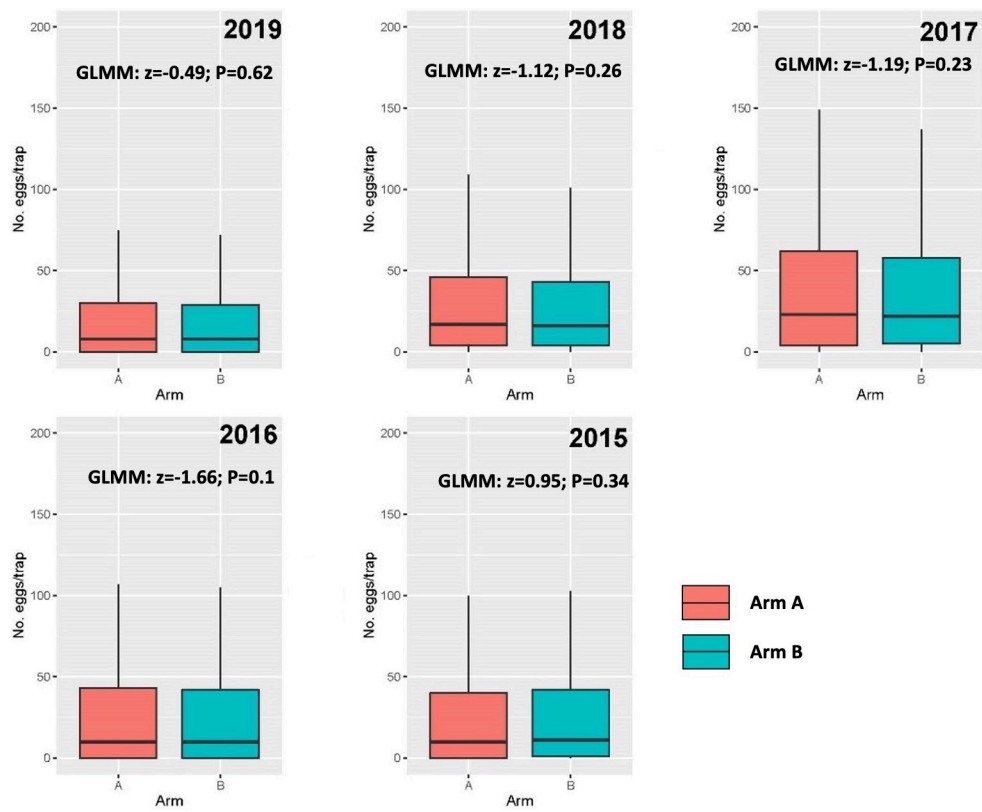

**Fig 5. Density of mosquitoes in the TIRS clusters.** Boxplots showing the Median (Q1-Q3, range) of the number of eggs per ovitrap per year (during the five years before the start of the study), stratified by the blinded control and intervention study arms (red for Arm A, blue for Arm B). Each box includes the z-score and P-value of the parameter 'Arm' in the GLMM evaluating the difference in the number of eggs per trap by study arm.

previously infected by ZIKV-alone with most potential DENV-exposed participants showing cross-reactive antibodies against ZIKV or other DENV serotypes.

According to WHO policy recommendation, the highest quality evidence for efficacy of a vector control intervention is generated by field trials measuring epidemiological endpoints [28]. Particularly for ABVs and urban vector control tools, the implementation of cluster-randomized controlled trials (CRCTs) has been limited to a few interventions, such as house screens, mosquito traps, spatial repellents, and *Wolbachia* biocontrol [20, 21]. Part of the challenge in designing an effective CRCT for ABVs emerges from the complex nature of virus transmission, which includes substantial heterogeneity in the distribution of *Ae. aegypti* within cities, a disproportionate contribution of inapparent infections to transmission, and the strong effect of human mobility modulating where and when exposure occurs [32]. We developed and applied an expanded covariate-constrained randomization algorithm to select clusters from within the hotspot region while maximizing the distance between clusters and minimizing imbalance [29]. We use a fried egg design and capture human mobility data to limit contamination and enable a range of analytical adjustments. Per protocol, only the statistical analysis team was unblinded to the identity of each trial arm. Once the trial is open for analysis, blinding will be lifted. For the purposes of this manuscript, the use of blinded trial arms allows depicting the aspects of the trial that were involved in participant enrollment and baseline information while respecting the blinding.

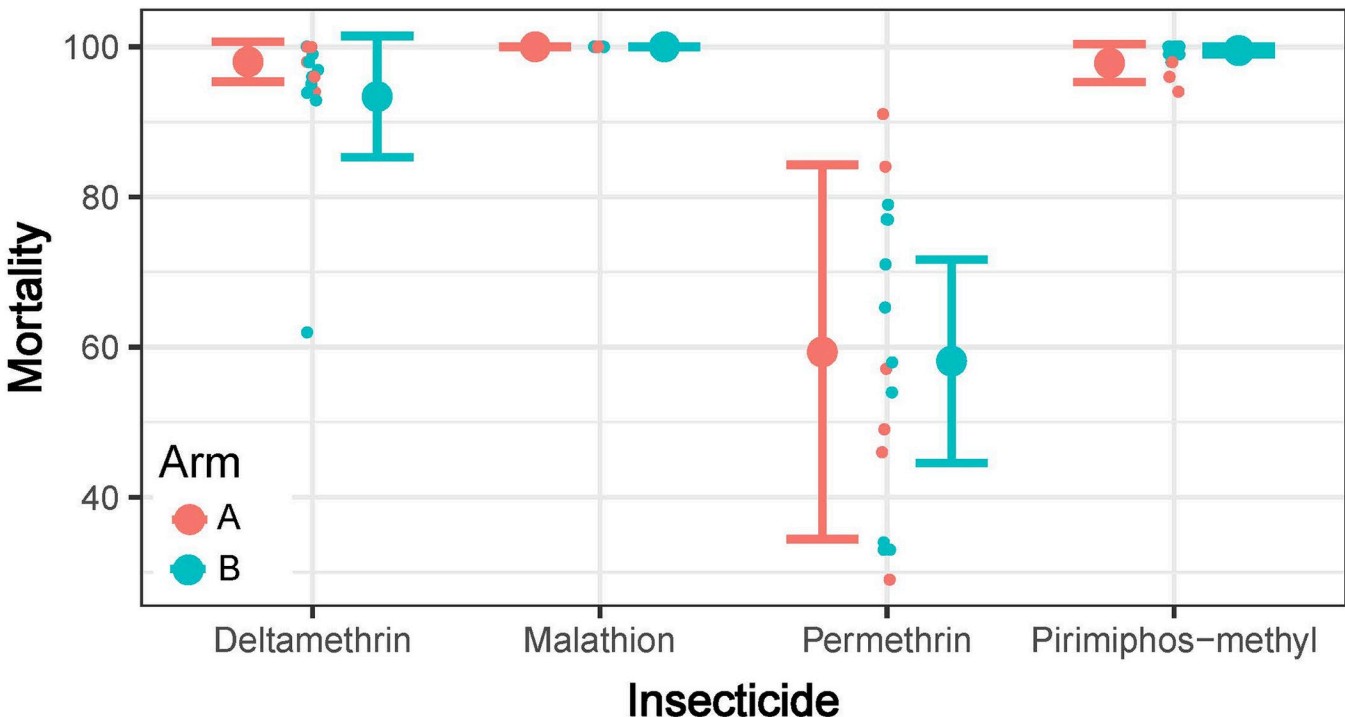

**Fig 6. Susceptibility of mosquitoes to insecticides.** Mortality (%) of *Ae. aegypti* exposed to different insecticide active ingredients using the CDC bottle bioassay. Dots indicate the mortality for each of the 16 clusters analyzed, colors differentiate between treatment and control clusters (blinded as Arm A and Arm B). Large dots and bars show the mean and 95% CI, respectively.

The COVID-19 pandemic represented an unprecedented disruption to livelihoods and economies. Merida was not isolated from the disruptions to travel, mobility, work, and social interaction, with the Ministry of Health implementing a packet of non-pharmacological interventions (e.g., social distancing, limited capacity in businesses, virtual school, and an alcohol sales ban) which significantly prevented transmission peaks in the early stages of the pandemic [41]. With social distancing being one of the most extensive measures (in terms of duration and adoption by the population), our baseline mobility data showed the strong trend of our entire cohort staying at home (more than 80% of the time) and the limited number and frequency of visits to other locations, primarily houses of relatives [41]. Social distancing and non-pharmacological interventions significantly impacted our trial, specifically participant enrollment and blood specimen procurement. By outlining specific plans and responsibilities for participant contact and follow-up, modifying activities to include outdoor contacts, and merging enrollment and phlebotomy activities, we minimized contact with participants and secured all required information during baseline. An unexpected outcome of COVID-19 was a large number of families moving out of their original cluster (or even outside Merida) due to economic or family issues, which meant our teams had to expand enrollment to up to 4,700 participants. Some practices imposed by COVID-19 (remote check-in with participants by phone calls or SMS) remained throughout the study and have increased the efficiency of detection and participant follow-up. Furthermore, by adopting a comprehensive set of safety measures (including PPE, minimal contact with participants and work schedules) we experienced no COVID-19 outbreak among TIRS trial personnel.

Sero-epidemiological studies of ABVs in Mexico are scarce, dispersed, restricted to local settings in selected states, focused on young age groups, based on diverse serological tests and

sources of human samples in various epidemiological settings (outbreak, transversal, cohort studies). Nevertheless, all studies point out the high transmission of DENV infection in every period analyzed, even in those under 25 years old [53–55]. In Yucatan, recent serological studies (2014–2015) in schoolchildren reported increasing seropositivity with age from 26.9% under five years old, 43.9% in 5 to 9 years old, and 61.4% in children 10 to 14 years old with different magnitudes in urban Merida [34]. Our seroprevalence estimates are similar to findings from that survey conducted five years prior, suggesting continual ABV transmission without a large outbreak within the period [52]. These conclusions are supported by the large proportion of participants with antibodies that neutralized ZIKV-alone (**Fig 4B**). Due to rapid and temporally isolated transmission of ZIKV during the 2016 outbreak in Mexico [56, 57], it is likely that many of these participants were exposed during that year and have likely not experienced a flavivirus exposure since then. The high proportion of TIRS participants that are DENV-naïve or exhibit monotypic DENV immunity indicates a high degree of susceptibility to future DENV infection in this cohort.

*Aedes aegypti* shows strong heterogeneity in its city-wide distribution [58], which could lead to the occurrence of ABV transmission hotspots [52]. To minimize potential bias due to the placement of clusters in neighborhoods with low *Ae. aegypti* density, the TIRS trial focused on an area identified as a hotspot of ABV transmission, which accounted for 50% of all cases reported in the city [52]. Within this area, *Ae. aegypti* larval stages are commonly found in small plastic containers and buckets, which generate a high productivity [48] and indoor adult density [37]. Our finding of a non-significant difference in indoor *Ae. aegypti* and outdoor ovitrap indices confirms that baseline entomological indicators are balanced between treatment arms. Despite detection of pyrethroid resistance in *Ae. aegypti*, the populations within the study clusters were susceptible to pirimiphos-methyl, the active ingredient of the insecticide formulation (Actellic 300CS, Syngenta) that will be used in the TIRS trial. The choice of insecticide was informed by a prior entomological CRCT conducted in Merida that showed that preventive TIRS using Actellic 300CS led to a significant reduction in *Ae. aegypti* for up to 7 months (the longest recorded for a residual insecticide controlling *Ae. aegypti* under field conditions) [23]. The TIRS trial will be uniquely positioned to quantify whether this meaningful entomological impact translates into a measurable public health benefit.

## Supporting information

**S1 Checklist. International Clinical Trials Registry Platform (ICTRP) checklist.**
(DOCX)

**S1 Fig. Flowchart of enrollment and loss to follow-up of the TIRS cohort.**
(TIF)

**S2 Fig. Representative FRNT curves and ages for single positive and indeterminate participants.** (**A**) FRNT curves and calculated FRNT$_{50}$ values for representatives of the indicated monotypic or multitypic exposure groups (left) against the virus indicated (top) for individual participants (participant ID [PID] indicated on left). (**B**) Ages of participants at time of sampling (median in red).
(TIF)

## Acknowledgments

The authors acknowledge Scott Ritchie for his inspiring contribution to the development of the TIRS methodology. Drs. Michael Dunbar, Gregor Devine, Richard Reithinger, Gabriela Gonzalez-Olvera, Wilbert Bibiano-Marin, and Tom Hladish for contributions to the design or

conceptualization of the trial. The authors acknowledge the serology laboratory at the SSY public health lab in Merida for their valuable technical support processing IgG ELISAs and the field team of the TIRS project including nurses, physicians, social workers, and bioinformaticians for their valuable contributions in this trial.

## Author Contributions

**Conceptualization:** James T. Earnest, Oscar D. Kirstein, Azael C. Mendoza, Gloria A. Barrera-Fuentes, Henry Puerta-Guardo, Matthew H. Collins, M. Elizabeth Halloran, Ira Longini, Natalie Dean, Lance A. Waller, Pablo Manrique-Saide, Gonzalo M. Vazquez-Prokopec.

**Data curation:** James T. Earnest, Oscar D. Kirstein, Azael C. Mendoza, Gloria A. Barrera-Fuentes, Henry Puerta-Guardo, Gabriela Gonzalez-Olvera, Anuar Medina-Barreiro, Audrey Lenhart, M. Elizabeth Halloran, Ira Longini, Natalie Dean, Lance A. Waller, Amy M. Crisp, Gonzalo M. Vazquez-Prokopec.

**Formal analysis:** Oscar D. Kirstein, Azael C. Mendoza, Gloria A. Barrera-Fuentes, Henry Puerta-Guardo, Matthew H. Collins, Gabriela Gonzalez-Olvera, Audrey Lenhart, M. Elizabeth Halloran, Ira Longini, Natalie Dean, Lance A. Waller, Amy M. Crisp, Pablo Manrique-Saide, Gonzalo M. Vazquez-Prokopec.

**Funding acquisition:** Matthew H. Collins, M. Elizabeth Halloran, Ira Longini, Natalie Dean, Lance A. Waller, Pablo Manrique-Saide, Gonzalo M. Vazquez-Prokopec.

**Investigation:** James T. Earnest, Oscar D. Kirstein, Azael C. Mendoza, Gloria A. Barrera-Fuentes, Henry Puerta-Guardo, Manuel Parra-Cardeña, Kevin Yam-Trujillo, Gabriela Gonzalez-Olvera, Audrey Lenhart.

**Methodology:** James T. Earnest, Oscar D. Kirstein, Azael C. Mendoza, Gloria A. Barrera-Fuentes, Henry Puerta-Guardo, Manuel Parra-Cardeña, Kevin Yam-Trujillo, Matthew H. Collins, Norma Pavia-Ruz, Guadalupe Ayora-Talavera, Gabriela Gonzalez-Olvera, M. Elizabeth Halloran, Ira Longini, Natalie Dean, Lance A. Waller, Amy M. Crisp, Fabian Correa-Morales, Jorge Palacio-Vargas, Pilar Granja-Perez, Salha Villanueva, Hugo Delfin-Gonzalez, Hector Gomez-Dantes, Pablo Manrique-Saide, Gonzalo M. Vazquez-Prokopec.

**Project administration:** James T. Earnest, Oscar D. Kirstein, Azael C. Mendoza, Gloria A. Barrera-Fuentes, Henry Puerta-Guardo, Matthew H. Collins, Norma Pavia-Ruz, Guadalupe Ayora-Talavera, Gabriela Gonzalez-Olvera, Anuar Medina-Barreiro, Wilberth Bibiano-Marin, M. Elizabeth Halloran, Ira Longini, Natalie Dean, Lance A. Waller, Fabian Correa-Morales, Jorge Palacio-Vargas, Pilar Granja-Perez, Salha Villanueva, Hugo Delfin-Gonzalez, Hector Gomez-Dantes, Pablo Manrique-Saide, Gonzalo M. Vazquez-Prokopec.

**Resources:** Gloria A. Barrera-Fuentes, Guadalupe Ayora-Talavera, Pilar Granja-Perez, Salha Villanueva, Hugo Delfin-Gonzalez, Hector Gomez-Dantes, Pablo Manrique-Saide, Gonzalo M. Vazquez-Prokopec.

**Supervision:** Oscar D. Kirstein, Azael C. Mendoza, Gloria A. Barrera-Fuentes, Henry Puerta-Guardo, Matthew H. Collins, Norma Pavia-Ruz, Guadalupe Ayora-Talavera, Gabriela Gonzalez-Olvera, Anuar Medina-Barreiro, Wilberth Bibiano-Marin, Audrey Lenhart, M. Elizabeth Halloran, Ira Longini, Natalie Dean, Lance A. Waller, Fabian Correa-Morales, Jorge Palacio-Vargas, Pilar Granja-Perez, Salha Villanueva, Hugo Delfin-Gonzalez, Hector Gomez-Dantes, Pablo Manrique-Saide, Gonzalo M. Vazquez-Prokopec.

**Validation:** James T. Earnest, Oscar D. Kirstein, Gloria A. Barrera-Fuentes, Henry Puerta-Guardo, Manuel Parra-Cardeña, Kevin Yam-Trujillo.

**Visualization:** Oscar D. Kirstein, Henry Puerta-Guardo.

**Writing – original draft:** James T. Earnest, Oscar D. Kirstein, Azael C. Mendoza, Gloria A. Barrera-Fuentes, Henry Puerta-Guardo, Gonzalo M. Vazquez-Prokopec.

**Writing – review & editing:** James T. Earnest, Oscar D. Kirstein, Azael C. Mendoza, Gloria A. Barrera-Fuentes, Henry Puerta-Guardo, Manuel Parra-Cardeña, Kevin Yam-Trujillo, Matthew H. Collins, Norma Pavia-Ruz, Guadalupe Ayora-Talavera, Gabriela Gonzalez-Olvera, Anuar Medina-Barreiro, Wilberth Bibiano-Marin, Audrey Lenhart, M. Elizabeth Halloran, Ira Longini, Natalie Dean, Lance A. Waller, Amy M. Crisp, Fabian Correa-Morales, Jorge Palacio-Vargas, Pilar Granja-Perez, Salha Villanueva, Hugo Delfin-Gonzalez, Hector Gomez-Dantes, Pablo Manrique-Saide, Gonzalo M. Vazquez-Prokopec.

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
