## [Decision Letter · Decision Letter 0]

22 Jul 2024

PONE-D-24-00764The TIRS trial: enrollment procedures and baseline characterization of a pediatric cohort to quantify the epidemiologic impact of targeted indoor residual spraying on Aedes-borne viruses in Merida, MexicoPLOS ONE

Dear Dr. Vazquez-Prokopec,

Thank you for submitting your manuscript to PLOS ONE. After careful consideration, we feel that it has merit but does not fully meet PLOS ONE’s publication criteria as it currently stands. Therefore, we invite you to submit a revised version of the manuscript that addresses the points raised during the review process. The title is quite self-explanatory: "The TIRS trial: enrollment procedures and baseline characterization of a pediatric cohort to quantify the epidemiological impact of targeted indoor residual spraying on Aedes-borne viruses in Merida, Mexico", is a well-written manuscript that aims to describe the selection criteria and profile of participants, being studied for a Targeted Indoor Residual Spraying (TRIS) program, whose objective is to reduce the transmission of arboviruses (DENV, CHIKV and ZIKV) in Mérida.

Small changes were suggested by the reviewers and once followed, I believe that the work will improve and will be ready to be published.

We look forward to receiving your revised manuscript.

Kind regards,

André Ricardo Ribas Freitas

Academic Editor

PLOS ONE

Journal Requirements:

2. In the online submission form, you indicated that "ata cannot be shared publicly because of confidentiality. Data are available from the corresponding author by email request."

**Additional Editor Comments:**

The title is quite self-explanatory: "The TIRS trial: enrollment procedures and baseline characterization of a pediatric cohort to quantify the epidemiological impact of targeted indoor residual spraying on Aedes-borne viruses in Merida, Mexico", is a well-written manuscript that aims to describe the selection criteria and profile of participants, being studied for a Targeted Indoor Residual Spraying (TRIS) program, whose objective is to reduce the transmission of arboviruses (DENV, CHIKV and ZIKV) in Mérida. As a sequential activity of the project, we await the results in new publications.

Reviewers' comments:

Reviewer's Responses to Questions

**Comments to the Author**

1. Is the manuscript technically sound, and do the data support the conclusions?

Reviewer #1: Yes

Reviewer #2: Yes

Reviewer #3: Yes

2. Has the statistical analysis been performed appropriately and rigorously? 

Reviewer #1: Yes

Reviewer #2: Yes

Reviewer #3: Yes

3. Have the authors made all data underlying the findings in their manuscript fully available?

Reviewer #1: No

Reviewer #2: Yes

Reviewer #3: Yes

4. Is the manuscript presented in an intelligible fashion and written in standard English?

Reviewer #1: Yes

Reviewer #2: Yes

Reviewer #3: Yes

5. Review Comments to the Author

Reviewer #1: This manuscript describes baseline finding for a large cRCT for a clinical trial evaluating Targeted Indoor Residual Spraying to reduce Aedes Borne Virus disease in Merida, Mexico. The objectives of the manuscript appear to have three objectives: 1) clearly describe the trial methodology so it can be cited in future publications (I support this approach), 2) demonstrate that clusters are well balanced on a large variety of relevant factors between treatment and control clusters prior to deployment of the intervention, and 3) provides additional insight about a trial that had to adjust in “real time” to the COVID-19 pandemic. Full disclosure, I reviewed an earlier version of this manuscript when submitted to PLOS NTD and can say the authors clearly considered my previous suggestions and concerns made at the time. Overall, I think this manuscript adds value to the literature on vector control trials for Aedes borne Viruses, demonstrating a very rigorous study design and approach. One could argue that this information should be included in a publication of the trial results, but I see value in this detailed presentation at the outset. It also provides some insight on ABV seroprevalence across Merida. I think this would be of interest to PLOS ONE readers and merits publication. Below are a few general and specific comments that the authors might consider.

General Comments:

Blinding. I understand why the authors have stratified their results into A and B to not reveal the treatment or control assignments which is appropriate. Some trialists (purists) might object to this, I personally do not, but I suggest the authors describe any blinding they are doing (this is some text already in the methods), but maybe a stronger statement in the discussion about the NEED and PRACTICALITY of blinding. So follow up questions associated with this are:

1) Why are seroprevalence results not presented for A and B arms. This is the most critical variable where you need to show balance and if not balance, rates would need to be age adjusted in the final primary outcome analyses. (This is Table 3 and Figure 3). Maybe this was an oversight but strange that this is the only data category not stratified.

2) Suggested Figure would be Age-seroprevalence curves (much in the text and Table 3, but this would be a contribution to the field).

Data Availability: I do not see any statement but imagine that the authors would be restricted from posting data at this stage. I’m not concerned but was a question posed by the journal.

Specific comments.

Make sure you check throughout Aedes aegypti or Ae. aegypti there are many places were Aegypti is capitalized.

Reviewer #2: The study design, statistical analysis are very rigorous. Results are clearly and completely presented and has only minor issues for revision:

"Ae Aegypti" Should be "Ae. aegypti"

Figure 3: quality is low, wich does not allow to understand it.

Figure 5: Mortality (%)

Reviewer #3: This is a well-written manuscript, which describes the recruitment and baseline characteristics of study participants for a Targeted Indoor Residual Spraying (TRIS) program, the goal of which is to reduce arbovirus (DENV, CHIKV, and ZIKV) transmission in Merida, Mexico. Baseline seroprevalence data and demographic information are provided. As a side note, the authors should be commended for keeping the study on track, despite the challenges faced due to the COVID-19 pandemic.

Line 46 (and elsewhere): replace “dengue (DENV), Zika (ZIKV), and chikungunya (CHIKV) virus” with “dengue virus (DENV), Zika virus (ZIKV), and chikungunya virus (CHIKV)”

Line 47: delete “ongoing”. It implies that the COVID-19 pandemic is still ongoing at this very moment in time, which isn’t really the case

Line 56: replace “seropositive for antibodies against one or more flaviviruses” with “seropositive for one or more flaviviruses”. Similar changes need to be made elsewhere. The word “antibodies” is redundant because “seropositive” means that antibodies were detected

Abstract: perhaps I’m being a bit picky with this one, but it would be helpful if it was explicitly stated in the abstract that indoor spraying has not yet commenced

Lines 83-84: it would be better to say “among the leading causes” because the global number of JEV or YFV cases may exceed that of ZIKV or CHIKV

Line 146: add a space between “(ICTRP)(see”

Lines 252-253: avoid using the word “genus” or replace “flavivirus” with “orthoflavivirus” throughout the manuscript. The genus Flavivirus has been renamed to Orthoflavivirus.

Line 264: is 72 h enough time for all dengue virus serotypes? Check that the wrong number hasn’t been added

Line 268: replace “mouse monoclonal” with mouse flavivirus-reactive monoclonal” or something similar

Tables 1 & 3: it would be helpful if the title of the table was above (rather than below) the table

Line 388: replace “IgG antibodies” with “IgG”. We all know that IgG is a type of antibody so it is redundant to include the word “antibodies”

Line 390: it says that seroprevalence is similar between both arms. It also says that the ratio of male and female participants is similar between both arms. The authors explain that this info is provided in Table 3. However, I don’t see “arm data” in the table, only “overall data”. Reword the text or clearly show the Arm A vs B data as done in Tables 1 and 2.

Table 4: arm data are not provided (only overall data)

6. PLOS authors have the option to publish the peer review history of their article (what does this mean?). If published, this will include your full peer review and any attached files.

Reviewer #1: No

Reviewer #2: No

Reviewer #3: No

---

## [Author Response · Author response to Decision Letter 0]

15 Aug 2024

Response to Reviewers’ Comments

Reviewer #1: This manuscript describes baseline finding for a large cRCT for a clinical trial evaluating Targeted Indoor Residual Spraying to reduce Aedes Borne Virus disease in Merida, Mexico. The objectives of the manuscript appear to have three objectives: 1) clearly describe the trial methodology so it can be cited in future publications (I support this approach), 2) demonstrate that clusters are well balanced on a large variety of relevant factors between treatment and control clusters prior to deployment of the intervention, and 3) provides additional insight about a trial that had to adjust in “real time” to the COVID-19 pandemic. Full disclosure, I reviewed an earlier version of this manuscript when submitted to PLOS NTD and can say the authors clearly considered my previous suggestions and concerns made at the time. Overall, I think this manuscript adds value to the literature on vector control trials for Aedes borne Viruses, demonstrating a very rigorous study design and approach. One could argue that this information should be included in a publication of the trial results, but I see value in this detailed presentation at the outset. It also provides some insight on ABV seroprevalence across Merida. I think this would be of interest to PLOS ONE readers and merits publication. Below are a few general and specific comments that the authors might consider.

We thank the reviewer for all of their effort in improving our manuscript as well as their kind words written here.

General Comments:

Blinding. I understand why the authors have stratified their results into A and B to not reveal the treatment or control assignments which is appropriate. Some trialists (purists) might object to this, I personally do not, but I suggest the authors describe any blinding they are doing (this is some text already in the methods), but maybe a stronger statement in the discussion about the NEED and PRACTICALITY of blinding. 

We added the following statement to the discussion “Per protocol, only the statistical analysis team was unblinded to the identity of each trial arm. Once the trial is open for analysis, blinding will be lifted. For the purposes of this manuscript, the use of blinded trial arms allows depicting the aspects of the trial that were involvement in participant enrollment and baseline information while respecting the blinding.” (lines 516-520) 

So follow up questions associated with this are:

1) Why are seroprevalence results not presented for A and B arms. This is the most critical variable where you need to show balance and if not balance, rates would need to be age adjusted in the final primary outcome analyses. (This is Table 3 and Figure 3). Maybe this was an oversight but strange that this is the only data category not stratified.

We have added arm data to table 3 and we respectfully argue that figure 3 (now fig 4) contains so much information that adding arm information would be confusing to readers (we have already addressed complaints that the data is hard to read in it’s current format).Similar to how the arms are balanced in terms of age, sex, and seroprevalence we did not observe any significant difference in FRNT titers between the arms. 

2) Suggested Figure would be Age-seroprevalence curves (much in the text and Table 3, but this would be a contribution to the field).

This figure is now Fig 3 in the manuscript

Data Availability: I do not see any statement but imagine that the authors would be restricted from posting data at this stage. I’m not concerned but was a question posed by the journal.

Correct, the data has been uploaded to Mendeley and will be released upon acceptance (see lines 317-319)

Specific comments.

Make sure you check throughout Aedes aegypti or Ae. aegypti there are many places were Aegypti is capitalized.

These mistakes have been corrected

Reviewer #2: The study design, statistical analysis are very rigorous. Results are clearly and completely presented and has only minor issues for revision:

We thank the reviewer for their kind words

"Ae Aegypti" Should be "Ae. aegypti"

These mistakes have been corrected

Figure 3: quality is low, wich does not allow to understand it.

We are not sure why this is the case, but it is an issue with the compiled version of the manuscript and not our submitted TIFF. Hopefully a more clear version will be included in this submission

Figure 5: Mortality (%)

I apologize, but it is unclear to me what this means. If there is a correction needed, please clarify.

Reviewer #3: This is a well-written manuscript, which describes the recruitment and baseline characteristics of study participants for a Targeted Indoor Residual Spraying (TRIS) program, the goal of which is to reduce arbovirus (DENV, CHIKV, and ZIKV) transmission in Merida, Mexico. Baseline seroprevalence data and demographic information are provided. As a side note, the authors should be commended for keeping the study on track, despite the challenges faced due to the COVID-19 pandemic.

We thank this reviewer for their work and kind words here.

Line 46 (and elsewhere): replace “dengue (DENV), Zika (ZIKV), and chikungunya (CHIKV) virus” with “dengue virus (DENV), Zika virus (ZIKV), and chikungunya virus (CHIKV)”

This has been corrected as suggested (lines 46-47)

Line 47: delete “ongoing”. It implies that the COVID-19 pandemic is still ongoing at this very moment in time, which isn’t really the case

This has been corrected as requested (line 49)

Line 56: replace “seropositive for antibodies against one or more flaviviruses” with “seropositive for one or more flaviviruses”. Similar changes need to be made elsewhere. The word “antibodies” is redundant because “seropositive” means that antibodies were detected

This change has been included (line 59)

Abstract: perhaps I’m being a bit picky with this one, but it would be helpful if it was explicitly stated in the abstract that indoor spraying has not yet commenced

We agree that this is important to include in the abstract, we have made revisions to the abstract to reflect that TIRS spraying has not yet occurred (lines 54-56)

Lines 83-84: it would be better to say “among the leading causes” because the global number of JEV or YFV cases may exceed that of ZIKV or CHIKV

This statement has been changed as requested (line 87)

Line 146: add a space between “(ICTRP)(see”

Fixed (line 149)

Lines 252-253: avoid using the word “genus” or replace “flavivirus” with “orthoflavivirus” throughout the manuscript. The genus Flavivirus has been renamed to Orthoflavivirus.

Thank you for pointing this out, we have changed the word to “orthoflavivirus” (line 257)

Line 264: is 72 h enough time for all dengue virus serotypes? Check that the wrong number hasn’t been added

In our hands, this assay works at 72h for the virus serotypes we used, though we are aware that different viruses require different incubation times.

Line 268: replace “mouse monoclonal” with mouse flavivirus-reactive monoclonal” or something similar

We have changed the wording to “mouse monoclonal antibody that is cross-reactive to multiple orthoflaviviruses” (line 274)

Tables 1 & 3: it would be helpful if the title of the table was above (rather than below) the table

We have changed the location of the table titles to above the tables per your suggestion and editorial guidelines

Line 388: replace “IgG antibodies” with “IgG”. We all know that IgG is a type of antibody so it is redundant to include the word “antibodies”

This change has been included (line 400)

Line 390: it says that seroprevalence is similar between both arms. It also says that the ratio of male and female participants is similar between both arms. The authors explain that this info is provided in Table 3. However, I don’t see “arm data” in the table, only “overall data”. Reword the text or clearly show the Arm A vs B data as done in Tables 1 and 2.

We have added arm information to a new version of table 3

Table 4: arm data are not provided (only overall data)

The table shows the result of a GLMM model comparing Arm A (baseline) with B. So, the betas and Pvalue are indicative of whether Arm B differed from Arm A. Here, ‘Arm’ is the treatment variable not an individual arm.

---

## [Editor Report · Decision Letter 1]

2 Sep 2024

The TIRS trial: enrollment procedures and baseline characterization of a pediatric cohort to quantify the epidemiologic impact of targeted indoor residual spraying on Aedes-borne viruses in Merida, Mexico

PONE-D-24-00764R1

Dear Dr. Vazquez-Prokopec,

We’re pleased to inform you that your manuscript has been judged scientifically suitable for publication and will be formally accepted for publication once it meets all outstanding technical requirements.

Kind regards,

André Ricardo Ribas Freitas

Academic Editor

PLOS ONE

Additional Editor Comments (optional):

I hope this message finds you well.

I am pleased to inform you that your manuscript titled "The TIRS trial: enrollment procedures and baseline characterization of a pediatric cohort to quantify the epidemiologic impact of targeted indoor residual spraying on Aedes-borne viruses in Merida, Mexico" has been accepted for publication in PLOS ONE.

We appreciate the thorough and thoughtful revisions you made in response to the reviewers' comments. The reviewers acknowledged the significant improvements, and we are confident that your article now meets the high standards required by our journal.

Your attention to addressing the reviewers' feedback, including the discussion on blinding, the addition of data in the tables, and the correction of technical details, has not only enhanced the clarity of your manuscript but also strengthened the presentation and interpretation of your findings.

Your work makes a valuable contribution to the literature on vector control and epidemiological studies, particularly given the additional challenges posed by the COVID-19 pandemic.

You will soon receive further instructions regarding the next steps for the publication of your article. If you have any questions or need additional assistance, please do not hesitate to reach out.

Congratulations on the acceptance of your manuscript, and thank you for choosing PLOS ONE to share your important work.

Best regards,

André Ribas Freitas

---

## [Editor Report · Acceptance letter]

6 Sep 2024

PONE-D-24-00764R1 

PLOS ONE

Dear Dr. Vazquez-Prokopec, 

I'm pleased to inform you that your manuscript has been deemed suitable for publication in PLOS ONE. Congratulations! Your manuscript is now being handed over to our production team.

Kind regards, 

on behalf of

Dr. André Ricardo Ribas Freitas 

Academic Editor

PLOS ONE